

# Switching spatial scale reveals dominance-dependent social foraging tactics in a wild primate

Alexander E.G. Lee[1,2,3] and Guy Cowlishaw[2]

[1] Department of Zoology, University of Oxford, Oxford, United Kingdom
[2] The Institute of Zoology, Zoological Society of London, London, United Kingdom
[3] Centre of Excellence in Biological Interactions, Department of Biological and Environmental Sciences, University of Jyväskylä, Jyväskylä, Finland

Corresponding author
Alexander E.G. Lee, allee@jyu.fi, aleex19@gmail.com

## ABSTRACT

When foraging in a social group, individuals are faced with the choice of sampling their environment directly or exploiting the discoveries of others. The evolutionary dynamics of this trade-off have been explored mathematically through the producer-scrounger game, which has highlighted socially exploitative behaviours as a major potential cost of group living. However, our understanding of the tight interplay that can exist between social dominance and scrounging behaviour is limited. To date, only two theoretical studies have explored this relationship systematically, demonstrating that because scrounging requires joining a competitor at a resource, it should become exclusive to high-ranking individuals when resources are monopolisable. In this study, we explore the predictions of this model through observations of the natural social foraging behaviour of a wild population of chacma baboons (*Papio ursinus*). We collected data through over 800 h of focal follows of 101 adults and juveniles across two troops over two 3-month periods. By recording over 7,900 social foraging decisions at two spatial scales we show that, when resources are large and economically indefensible, the joining behaviour required for scrounging can occur across all social ranks. When, in contrast, dominant individuals can aggressively appropriate a resource, such joining behaviour becomes increasingly difficult to employ with decreasing social rank because adult individuals can only join others lower ranking than themselves. Our study supports theoretical predictions and highlights potentially important individual constraints on the ability of individuals of low social rank to use social information, driven by competition with dominant conspecifics over monopolisable resources.

## INTRODUCTION

Socially exploitative behaviours occur when individuals make use of the resources of competitors. A wide range of both theoretical and empirical studies over recent decades have highlighted such behaviours as a major potential cost of group living (*Giraldeau & Dubois, 2008*). Because resources such as food, mates, breeding territories, or safety from predation generally show variation in their distribution through space or time,

individuals should benefit from gathering information about their local environment to improve decision-making (*Valone, 1989*; *Valone, 2006*; *McNamara, Green & Olsson, 2006*). However, when the personal collection of information requires search effort or risk taking, selection should favour the avoidance of these costs through the collection and use of social information, where individuals attend to the behaviours of others in a social group to exploit their efforts and knowledge (for review, see *Valone & Templeton, 2002*; *Danchin et al., 2004*; *Rieucau & Giraldeau, 2011*).

The dynamics of these interactions have been formalised as the producer-scrounger game (*Barnard & Sibly, 1981*; *Barnard, 1984*; *Vickery et al., 1991*). Supported by a wide range of empirical studies (e.g., *Koops & Giraldeau, 1996*; *Mottley & Giraldeau, 2000*; *Morand-Ferron, Giraldeau & Lefebvre, 2007*), the producer-scrounger game has emerged as the prevailing theoretical framework in which to study social foraging decisions (*Vickery et al., 1991*; *Giraldeau & Caraco, 2000*). In this game, producers actively search for resources, while scroungers instead rely on social information to exploit the discoveries of producers. The two tactics are considered mutually exclusive. Scrounging is thus under negative frequency-dependent selection since its success, being dependent on the efforts of producers, is determined by the relative frequencies of the two tactics within a group. This dynamic is expected to lead populations to an evolutionarily or behaviourally stable mix of producing and scrounging (*Giraldeau & Dubois, 2008*; *Fawcett, Hamblin & Giraldeau, 2013*). As such, scrounging behaviour has the potential to reduce the per capita rate of discovery of new resources (*Vickery et al., 1991*), which may act to reduce average individual fitness in a population (*Coolen, Giraldeau & Vickery, 2007*).

The basic producer-scrounger model assumes that an individual's phenotype has no influence on its decision or ability to play either tactic. All individuals are essentially equivalent, and are expected to receive equal payoffs. However, many empirical studies have shown that an individual's tactic choice may be strongly influenced or constrained by its phenotype (e.g., *Beauchamp, 2001*; *Stahl et al., 2001*; *Di Bitetti & Janson, 2001*; *Kurvers et al., 2010*). This has potentially important fitness implications, since theory predicts that phenotype-limited games may not reach an evolutionarily stable mix of strategies, resulting in differential payoffs across individuals (*Parker, 1982*).

Since scrounging behaviour represents the exploitation of another's resource, one might expect it to be strongly influenced by social dominance. Specifically, the competitive advantage of high-ranking individuals should allow them to scrounge from others more easily (*Parker, 1974*; *Maynard Smith & Parker, 1976*; *Hammerstein, 1981*). Despite this expectation, empirical studies have not been unanimous. While some experiments have demonstrated a clear positive relationship between social dominance and scrounging behaviour (*Stahl et al., 2001*; *Liker & Barta, 2002*; *Lendvai, Liker & Barta, 2006*; *McCormack, Jablonski & Brown, 2007*), a number of other studies have not (*Bugnyar & Kotrschal, 2002*; *Robinette Ha & Ha, 2003*; *Beauchamp, 2006*; *Teichroeb, White & Chapman, 2015*). This conflict might be reconciled by considering more systematically the spatiotemporal distribution of resources faced by different taxa in both naturalistic and experimental settings. The competitive benefits of social dominance are expected to be associated with priority of access to resources, manifest as contest competition

(*Kaufmann, 1983*; *Łomnicki, 2009*). Consistent with this, resource defence theory predicts that individuals should be more aggressive when defending a resource in accordance with both its value and how easily it can be defended (*Grant, 1993*; *Grant & Guha, 1993*; *Robb & Grant, 1998*). Empirical studies into dominance and resource defence have demonstrated higher foraging success for socially dominant individuals only when presented with limited food patches that are monopolisable (*Theimer, 1987*; *Vahl et al., 2005*).

Some researchers have suggested that the integration of producer-scrounger and resource defence theory might elucidate an interesting relationship between socially exploitative behaviour and dominance (*Barta & Giraldeau, 1998*; *Giraldeau & Dubois, 2008*). Specifically, dominant individuals should benefit disproportionately if they can use their competitive advantage to ensure that only they can use social information effectively. Two studies (*Barta & Giraldeau, 1998*; *Lee et al., 2016*) have explored this hypothesis by modelling the effects of between-individual asymmetries in competitive ability on producer-scrounger dynamics in a group. They found that when social rank conferred no competitive advantage to an individual—that is, resources were not monopolisable— groups converged on basic producer-scrounger equilibria in which all individuals behave equivalently and receive equal payoffs. In contrast, when individuals could use their social rank to gain a competitive advantage in monopolising a resource, scrounging behaviour was strongly associated with dominance, and dominant individuals achieved the highest payoffs (*Barta & Giraldeau, 1998*; *Lee et al., 2016*). The driving force behind this pattern was the fact that scrounging behaviour requires that a competitor is joined at a resource in space and time, forging a causal link between the degree to which contest competition acts and constraints on an individual's ability to use social information (*Lee et al., 2016*). However, to date there has been no attempt to test these predictions empirically, either in the laboratory or under naturalistic conditions.

In this study, we explore a key prediction generated by the unification of producer-scrounger and resource defence theories, namely that there should be a strong link between social dominance and the scrounger tactic only when resources are monopolisable. We did this by studying the natural social foraging decisions made by wild chacma baboons (*Papio ursinus*) across two spatial scales that are expected to differ in the degree to which dominant individuals can monopolise food. At the first spatial scale—the 'patch'—resource clumps were too large to be monopolised independently, while at the second—the 'sub-patch'—resource clumps were smaller and monopolisation was possible (see 'Methods' for further details on how these spatial scales were defined). Because our focus was on naturalistic behaviour, we did not manipulate the information available to individuals while foraging to manufacture a situation where joining a competitor always represented the exclusive use of social information, which could accurately be termed scrounging (*Vickery et al., 1991*). Rather, we consider the observable joining behaviours of individuals as they foraged, representing competitive interactions fundamental to the predictions of producer-scrounger theory (*Lee et al., 2016*). In this way, we explore the competitive constraints that social rank may impose on an individual's ability to use social information through joining behaviour, and the relation of such constraints to resource monopolisability in a naturalistic setting in which information use is expected to be important. We make

the following three predictions: (1) joining behaviour should show a strong positive association with dominance rank at the smaller, sub-patch level but not at the larger, patch level; (2) individuals should join those to whom they are dominant at the sub-patch level, but there should be no systematic asymmetry in dominance when joining occurs at the patch level; and (3) joining should be associated with competitive exclusion (i.e., resource monopolisation) at the sub-patch level, but not at the patch level.

## MATERIALS AND METHODS

### Study site and species

Fieldwork was conducted at Tsaobis Nature Park, Namibia (22°23′S, 15°45′E), during two three-month periods between August and October in 2012 and 2013. Two groups of chacma baboons, hereafter referred to as troop 'J' (group size and compositions: $N_{J,2012} = 54$, adult females = 16, adult males = 15, juveniles = 23; $N_{J,2013} = 58$, adult females = 18, adult males = 9, juveniles = 31) and troop 'L' ($N_{L,2012} = 51$, adult females = 18, adult males = 6, juveniles = 27; $N_{L,2013} = 62$, adult females = 19, adult males = 11, juveniles = 32), were the focus of study. All baboons were individually recognisable and habituated to the presence of observers at close proximity. Each group was followed daily from dawn until dusk (see *Huchard et al., 2009* for further information). For each year, data were collected for all individual baboons >6 months of age (the age at which young baboons begin to forage independently of their mother) at the start of the study period, resulting in a total sample of 101 individuals (2012: 54 adults, 43 juveniles; 2013: 50 adults, 41 juveniles). Differences in the sample of individuals across years were due to death, emigration, or passing the minimum age threshold.

Chacma baboons are an ideal model system for our study, since they live in large, stable social groups in which linear dominance hierarchies are clear (*Altmann & Altmann, 1973*), individuals generally feed at the same time (*King & Cowlishaw, 2009*), the use of social information while foraging has been demonstrated in field-based experiments (*Carter, Torrents Ticó & Cowlishaw, 2016*), and socially exploitative foraging interactions are common (*King, Isaac & Cowlishaw, 2009*; *Marshall et al., 2012*). Furthermore, our study troops spent approximately 80% of their foraging time during the study period in a riparian woodland environment, characterised by large trees including *Faidherbia albida*, *Salvadora persica*, *Acacia erioloba*, *Acacia tortilis*, and *Prosopis glandulosa*. Within this feeding environment, we defined two spatial scales between which the ability of dominant individuals to monopolise food were predicted to differ: the patch and the sub-patch.

The patch represents the scale traditionally used in foraging theory and ecology, and is defined as a spatially discrete unit of a food resource (*Wiens, 1976*). Here, we refined this definition such that the operational definition of a patch was a single tree or shrub, or a collection of conspecific trees or shrubs growing together with a continuous canopy separated by no more than 1 m (median surface area = 156 m$^2$, interquartile range = 28–237 m$^2$; $n = 59$; see *Marshall et al., 2012* for further details). In contrast, the sub-patch was defined as the area in a patch within which an individual could feed without travelling (i.e., within arm's reach of a stationary baboon, approximately 2.25 m$^2$). This is equivalent

to the 'feeding station' scale that has received some attention in the foraging literature (*Kotliar & Wiens, 1990*; see *Searle, Hobbs & Shipley, 2005* for a review). Given the large size of patches compared with sub-patches, dominant individuals should be able to competitively exclude subordinate others more easily at the latter scale.

## Data collection and processing

Information regarding individual social foraging decisions and interactions at each spatial scale was recorded through focal sampling (*Altmann, 1974*) on Motorola ES400 Personal Digital Assistants and Google Nexus 4 Smartphones using a customised data capture application in the database-driven software Cybertracker v.3.317 (http://cybertracker.org). Focal follows lasted between 15 and 30 min, and the same individual was not studied more than once within a 6-hour period. Individuals were selected for focal observation using a pseudorandom sampling process, which ensured even coverage across different times of day (based on four consecutive 3-hour time blocks from 06:00 to 18:00) and different months.

A patch entry event was recorded whenever the focal individual searched for or consumed food in a new patch for 5 s or more. While in a patch, the focal individual could move between sub-patches. A sub-patch entry was recorded when an individual relocated into a new area of a patch to forage, and either remained stationary for $\geq$5 s while standing, or sat for $\geq$1 s, to forage in this location. In this way, foraging behaviour at each spatial scale was studied at the level of investment, since entries need not have resulted in successful food consumption (although in almost all cases did). At each spatial scale, a specific foraging decision was assigned to every entry event. The decision was defined as 'produce' if the patch or sub-patch being entered was unoccupied, and 'join' if occupied by a conspecific. Note that join events at the sub-patch level need not have been preceded by a join event at the patch level, because (1) a focal individual could enter an unoccupied patch and subsequently be joined by others, providing opportunities for future join events at the sub-patch level; and (2) focal follows could begin with the focal individual already occupying a patch.

For each join event, the number and identity of individuals occupying the resource was recorded. In cases where visibility was poor, a minimum number of occupants was estimated and, where known, their identity recorded. Since individuals being joined could either remain in, or be supplanted from, their patch or sub-patch, we recorded whether or not a join event was associated with competitive exclusion. We defined supplanting, representing competitive exclusion, as an approach-retreat interaction (*Rowell, 1966*; *Silk et al., 2010*) at a given patch or sub-patch that resulted in the entry and exit of the approaching and retreating individuals, respectively.

Since the size of a patch is variable, while sub-patch size is fixed, the relationship between them is such that at the smallest, or 'critical', patch sizes they reach equivalence. With this in mind, social foraging decisions were included only where a sub-patch structure could be defined (i.e., where the occupied patch held more than one sub-patch), such that the sub-patch always represented a smaller spatial scale nested within the patch. Study at the sub-patch scale thus captured social foraging dynamics at a resolution higher than at the

patch scale, allowing us to avoid conflating processes working at the two different spatial scales. Specifically, monopolisation of food at the patch scale always required the defence of an area at least (but generally considerably more than) double that required at the sub-patch scale. The data were then filtered further to exclude all ambiguous foraging decisions that could not clearly be classified as either produce or join (<10%). A total of 801 focal hours were carried out across the two study periods on 101 individual baboons (mean $\pm$ s.e. $= 7.9 \pm 0.1$ h per individual), resulting in a dataset of 1861 patch entry and 5050 sub-patch entry decisions for analysis. All observers completed a period of intensive training in the field to ensure high levels of accuracy and consistency in recognising patch and sub-patch boundaries, entry events, foraging decisions, and competitive exclusion. Observers were also naïve to the predictions of the study relating to associations between joining behaviour and social dominance at the two spatial scales.

A dominance hierarchy was generated for each troop-year combination using pairwise agonistic interactions occurring within each study period. These interactions were collected both during focal follows and through ad libitum sampling, and were used to make actor-receiver matrices indicating the number of agonistic interactions occurring between each dyad in each direction. No dominance interactions occurring during foraging decisions were included in the matrices. In addition, all interactions involving individuals not yet weaned from their mother were excluded, because dominance asymmetry at this age is strongly influenced by the mother's presence and behaviour (*Cheney, 1977*). Each actor-receiver matrix ($N_{2012,J} = 1010$; $N_{2012,L} = 1{,}025$; $N_{2013,J} = 833$; $N_{2013,L} = 1{,}073$) was reordered using Matman 1.1.4 (Noldus Information Technology 2003), optimised by selecting the hierarchy with the lowest level of conflict (i.e., minimising the number of interactions inconsistent with the predicted hierarchy) using a heuristic search algorithm with ten thousand randomisations. Linearity was supported for all four hierarchies (Landau's corrected linearity index: $h'_{2012,J} = 0.19$; $h'_{2012,L} = 0.32$; $h'_{2013,J} = 0.18$; $h'_{2013,L} = 0.15$, $p < 0.001$ in all cases), highlighting the rarity of interactions inconsistent with the predicted hierarchy ($n_{2012,J} = 67$ (7%); $n_{2012,L} = 81$ (8%); $n_{2013,J} = 43$ (5%); $n_{2013,L} = 71$ (7%)). Individuals not yet weaned were then re-entered into the appropriate dominance hierarchy based on their maternal rank (i.e., one position below their mother, consistent with the well-documented maternal reinforcement of offspring rank in chacma baboons; *Cheney, 1977*), producing complete hierarchies that included all members of the group for each year. To control for differences in the size of groups within and across years, all absolute ranks (ranging from 1 to $n$) were standardised to between 0 (lowest rank) and 1 (highest rank) following $1 - ((1 - r)/(1 - n))$, where $r$ is the absolute rank of an individual.

Our wholly observational research adhered to the Guidelines for the Use of Animals in Behavioural Research and Teaching (Animal Behaviour 2012 83:301–309), and our protocols were assessed and approved by the Ethics Committee of the Zoological Society of London (BPE/0518). Our study was approved by the Ministry of Environment and Tourism in Namibia (Research Permits 1696/2012 and 1786/2013).

## Statistical analyses

Our analysis was divided into three sections consistent with the three main predictions outlined above. First, we used generalised linear mixed-effects modelling (GLMM) to explore how the relationship between social dominance and joining behaviour changed across spatial scales due to differences in resource monopolisability. Our main prediction was that all individuals would exhibit joining behaviour at the patch scale, but that there would be a strong positive relationship between rank and joining at the sub-patch scale. However, since juvenile baboons are often tolerated at feeding sites (e.g., *Huchard et al., 2013*), we predicted that this positive relationship (and thus an interaction between spatial scale and dominance rank) would only hold for adults. While differences between adults and juveniles were not the focus of this study, it was important to include age class in our statistical models to fully understand any relationships between resource monopolisability, contest competition, and joining behaviour. We thus constructed our statistical model with a three-way interaction between spatial scale ('patch' or 'sub-patch'), dominance rank, and age class ('juvenile' or 'adult'). The response variable was given as a binary indicator of the decision at each entry to either 'produce' or 'join', scored as 0 or 1, respectively. We fit a binomial error structure to the GLMMs. Model selection was conducted by using a likelihood ratio test ($\alpha = 0.05$) to judge whether the model with or without the three-way interaction term provided the better fit to the data, and if the latter, whether those models with or without two-way interactions between spatial scale, dominance rank, and age class provided the better fit. Troop and year were included as control fixed effects, and were thus retained in all models. Focal identity and focal follow number were included as random intercepts in all models.

Second, we asked whether join 'events' were consistently associated with asymmetries in social rank at each spatial scale. Our main prediction was that individuals would consistently join others lower ranked than themselves at the sub-patch level, but would join others regardless of rank differences at the patch level. Again, we predicted that the relationship at the sub-patch level would not hold for juvenile individuals. To test this second set of predictions, we used a randomisation method to compare the joining behaviour we observed to the patterns of joining that would be expected if individuals joined others randomly with respect to rank difference. We employed this method because any relationship between dominance rank and joining frequency demonstrated in our first analysis would indicate that a crude comparison of the rank difference between joining and joined individuals could lead us to erroneous conclusions. For example, if high ranked individuals joined more frequently than low ranked individuals at a given spatial scale, our data would suggest that individuals on average joined others lower ranked than themselves at this spatial scale in the case that their actual joining behaviour was random with respect to rank difference, simply because individuals with above average social rank necessarily have more individuals subordinate to them than dominant to them. Our observation variable for this analysis was a binary indicator of whether the joining individual was dominant or subordinate to the joined individual. For those events where multiple individuals were joined in a patch (36%) or sub-patch (2%), the direction of their average rank difference with the focal individual was used. We generated expectations of rank differences under

random joining behaviour with respect to rank difference by randomly resampling from the appropriate troop only the identity of the joined individual for each observed join event, and calculating the difference in rank between the actual joiner and this randomly sampled individual. We repeated this process 10,000 times to generate a distribution of the expected proportion of join events in which the joining individual would be subordinate to the joined individual if individuals joined randomly with respect to rank difference. We defined that our observed estimates for the proportion of events with a subordinate joiner deviated from random expectations when they fell outside of the 95% tolerance intervals of the random distribution. We built four sets of random distributions to which we could compare our observational estimates: one for adult joiners and one for juvenile joiners at both the patch and the sub-patch scale.

Third, we built a GLMM to establish whether join events at different spatial scales were associated with differences in the competitive exclusion experienced by the joined individual. Competitive exclusion was modelled as a binary response variable: individuals were either supplanted from the resource or were not. Fixed effects were included as an interaction between spatial scale and age class, and were assessed using likelihood ratio tests as described above. Since these data were not available for patch level decisions in 2012, only decisions from 2013 were used in this analysis. Troop was included as a control fixed effect, and so was retained in all models, and focal identity and focal follow number were included as random intercepts. We predicted that joining behaviour in adults would cause competitive exclusion of the joined individual at the sub-patch but not the patch scale, and that joining behaviour in juveniles would result in lower levels of competitive exclusion at the sub-patch scale compared with adults.

All analyses were conducted in R version 3.0.2. using the lme4 package (*Bates et al., 2013*; *R Core Team, 2013*).

## RESULTS

### Dominance and social foraging decisions at different spatial scales

At the patch scale, joining behaviour was common regardless of social rank (Table 1; Fig. 1A), consistent with our predictions. Although there was some increase in joining with social dominance in adults, even the lowest ranked individuals entered occupied patches approximately 55% (95% confidence intervals: 41% and 67%) of the time. While joining was in general much less common at the sub-patch scale, there was a strong positive relationship in adults between dominance and joining behaviour (twice that at the patch scale) that was consistent with our predictions (Table 1; Fig. 1B). The lowest ranked adults had around a 1% probability (95% confidence intervals: 0% and 2%) of joining when entering a new sub-patch, while mid-ranked and top-ranked adults did so approximately 4% (95% confidence intervals: 3% and 6%) and 11% (95% confidence intervals: 8% and 17%) of the time, respectively.

As predicted, the effects of social rank on the probability of joining were weaker in juveniles, and this held across both spatial scales such that there was in general no relationship between dominance and joining frequency in juveniles (Table 1; Fig. 1). In

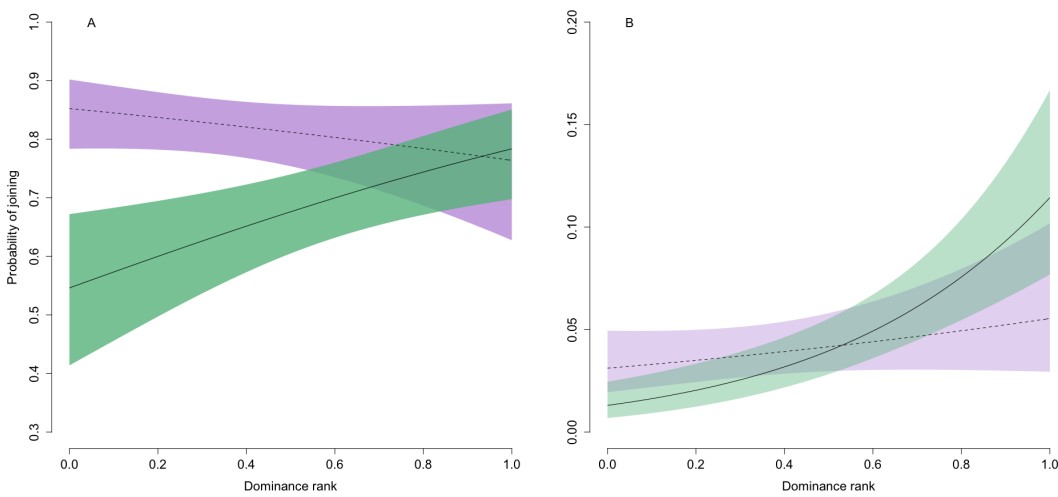

**Figure 1 Predicted relationship between the probability of joining behaviour and dominance rank at the (A) patch and (B) sub-patch level.** For both panels, solid and dashed lines represent the predicted values for adults and juveniles, respectively. Shaded green (adults) and purple (juveniles) regions are bounded by upper and lower 95% confidence intervals. Note the difference in scale of the two $y$-axes, reflecting the much lower levels of joining across all individuals at the sub-patch compared with patch scale.

**Table 1 Factors predicting the probability of joining behaviour and competitive exclusion associated with joining.**

| Response | N | Fixed effect | $\beta$ | s.e. | $\chi^2$ | $p$ |
|---|---|---|---|---|---|---|
| Probability of joining | 6,911 | Intercept | −0.05 | 0.26 | | |
| | | Spatial scale (Sub-patch) | −4.51 | 0.28 | | |
| | | Rank | 1.10 | 0.38 | | |
| | | Age class (Juvenile) | 1.57 | 0.30 | | |
| | | Troop (L) | 0.20 | 0.15 | | |
| | | Year (2013) | 0.27 | 0.12 | | |
| | | Spatial scale (Sub-patch) * Rank | 1.18 | 0.37 | 10.30 | 0.001 |
| | | Rank * Age class (Juvenile) | −1.68 | 0.54 | 9.03 | 0.003 |
| | | Spatial scale (Sub-patch) * Age class (Juvenile) | −0.68 | 0.21 | 10.40 | 0.001 |
| Probability of competitive exclusion | 385 | Intercept | −2.44 | 0.47 | | |
| | | Spatial scale (Sub-patch) | 4.55 | 0.65 | 164.27 | <0.001 |
| | | Age class (Juvenile) | −1.61 | 0.47 | 14.06 | <0.001 |
| | | Troop | −0.89 | 0.44 | | |

**Notes.**
Model reference categories: Spatial scale (Patch), Age class (Adult), Troop (J), Year (2012).

addition, juveniles were on average more likely than adults to join at the patch scale, but this primarily reflected low-ranking juveniles joining much more frequently than similarly ranked adults when entering a new patch. This effect was somewhat weakened at the sub-patch scale: low-ranked juveniles joined more frequently when entering a new sub-patch than similarly ranked adults but the pattern was reversed for high-ranked juveniles.

Combined, our results reflect support for the three possible two-way interactions between spatial scale, dominance rank, and age class (Table 1). Our lack of support for a three-way interaction between these effects (likelihood ratio test: $\chi^2 = 0.70$, $p = 0.41$) reflects the fact that there was some increase in joining with social rank in adults at both spatial scales. This meant that a stronger effect of dominance in adults at the sub-patch versus patch level, combined with no clear relationship between joining frequency and social rank in juveniles at either spatial scale, was captured by a general (i.e., not age class-specific) increase in joining with dominance at the sub-patch level and a general (i.e., not spatial scale-specific) decrease in the effect of dominance on joining in juveniles.

## Social constraints on joining behaviour at different spatial scales

At the patch scale, there was no evidence that adult individuals joined others systematically higher or lower ranked than themselves compared with the rank asymmetries expected if individuals joined others randomly with respect to their rank difference (Fig. 2A, $N_{patch,adult} = 313$). Although adults were less likely to join individuals dominant to them at this spatial scale, this could be explained under joining behaviour that was random with respect to rank difference by the finding in our first analysis that the frequency of joining behaviour increased slightly with increasing rank (Fig. 1A).

At the sub-patch scale, adult individuals joined individuals dominant to themselves in 9% of join events, and so were less likely to do so than would have been expected if individuals joined others randomly with respect to rank difference (Fig. 2B; $N_{sub\text{-}patch, adult} = 154$) and given the fact that high ranked individuals are much more likely than low ranked individuals to join others at this spatial scale (Fig. 1B). We thus found support for our prediction that adult individuals would consistently join others lower ranked than themselves at the sub-patch level.

Comparisons between observed and random joining behaviour for juveniles were broadly similar to those for adults. At the patch scale, the rank asymmetries at observed join events did not deviate from expectations under joining behaviour random with respect to rank difference (Fig. 2C; $N_{patch,juvenile} = 349$). At the sub-patch scale, we observed juveniles joining others to whom they were subordinate in 49% of cases. Although this shows that juveniles were frequently able to join others dominant to themselves, they nonetheless did so less often than expected under random joining (Fig. 2D; $N_{sub\text{-}patch, juvenile} = 147$). Despite the fact that our first analysis found no relationship between dominance and joining frequency for juveniles at the sub-patch level (Fig. 1B), the random distribution at this spatial scale reflects the fact that juveniles are usually below average in rank.

## Competitive exclusion at different spatial scales

Joining caused competitive exclusion at the sub-patch scale much more than it did at the patch scale (Table 1). In adults, joining at the patch scale was associated with competitive exclusion in 9% of cases. This figure increased to 79% at the sub-patch scale. Joining by juveniles was less likely to result in competitive exclusion at both spatial scales (patch: 3%; sub-patch: 51%).
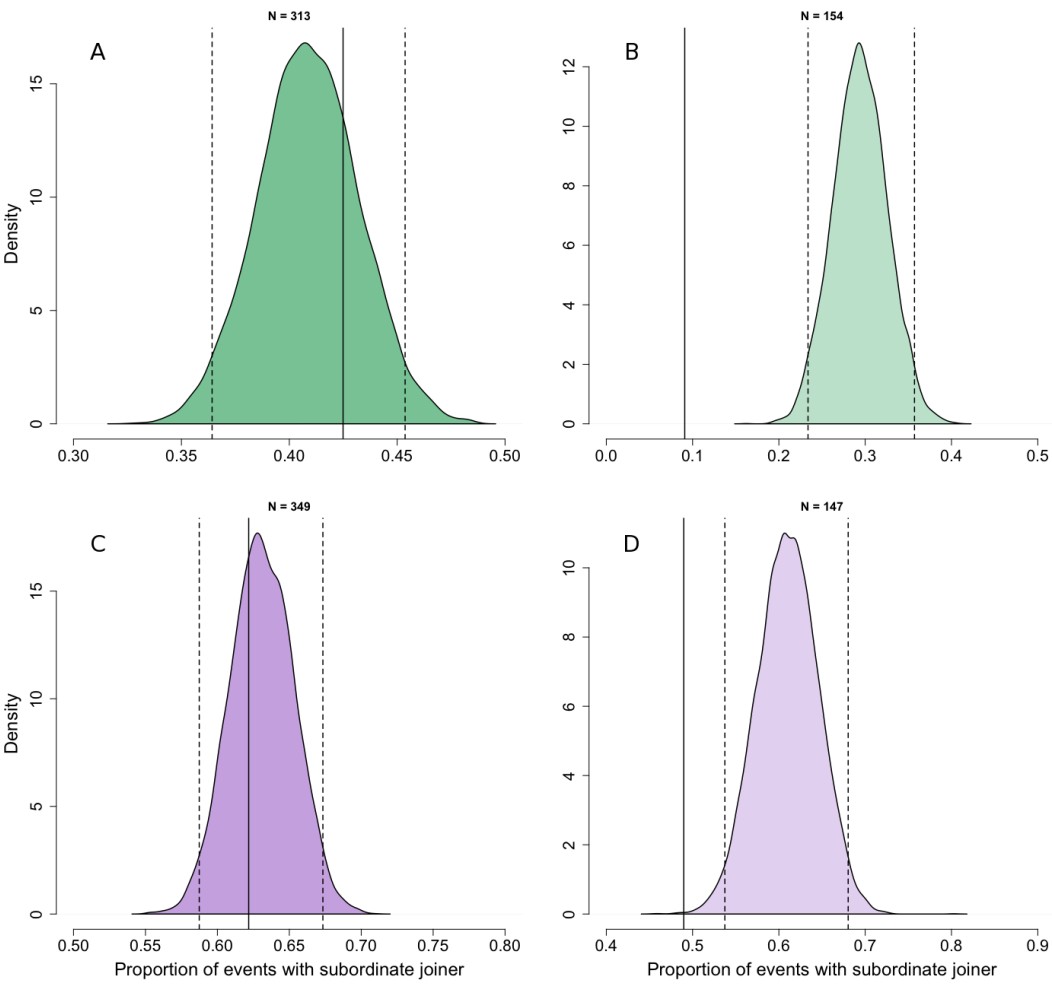

**Figure 2  Comparison of observed dominance asymmetry during join events with simulated joining behaviour that is random with respect to rank difference.** Probability density distributions show expectations for the proportion of join events in which the joining individuals would be subordinate to the joined individual if their behaviour was random with respect to the rank of the joined individual. The distributions are for adults (A) and juveniles (B) at the patch level, and adults (C) and juveniles (D) at the sub-patch level, generated through 10,000 iterations of randomly selecting the individual to be joined at each join event. For each distribution, dotted vertical lines indicate the 95% tolerance intervals and solid vertical lines indicate our observed value.

## DISCUSSION

We provide empirical evidence that joining behaviour should be more strongly related to social rank when the competitive asymmetries associated with dominance are stronger, in support of previous theoretical predictions (*Barta & Giraldeau, 1998*; *Lee et al., 2016*). We show that changes in resource monopolisability can mediate this shift in competitive asymmetry through changes in competitive exclusion at different spatial scales. At the larger patch scale, adults could join others regardless of any differences in their social ranks, and did so frequently. At the smaller sub-patch scale, joining was a rarer event— likely reflecting the higher finder's share at this spatial scale (*Vickery et al., 1991*)—but also

represented a more exclusive tactic, almost non-existent in the lowest-ranking individuals but increasing in probability with social dominance. When resources are monopolisable, socially subordinate individuals may thus be constrained in their ability to exploit available social information when its use requires joining behaviour, as is assumed by producer-scrounger theory. Since the size of a resource is expected to influence its economic defensibility (*Grant, 1993*), our study supports calls to unify producer-scrounger and resource defence theory in order to better understand the relationship between dominance and socially exploitative behaviours (*Barta & Giraldeau, 1998*; *Dubois, Giraldeau & Grant, 2003*; *Dubois & Giraldeau, 2005*; *Giraldeau & Dubois, 2008*). However, our study also highlights two areas in which our understanding of the evolutionary ecology of animal information use is still lacking, requiring further theoretical and empirical developments.

First, we showed a strong increase in the frequency of joining at higher ranks for adults at the sub-patch scale, where single individuals could use their dominance to exclude competitors. However, the pattern we observed was weaker than that predicted by *Barta & Giraldeau (1998)*, who suggested a complete absence of scrounging in all but the most dominant individuals when resources could be effectively monopolised. We found that even middle- and low-ranking adults can scrounge from others, provided that the others they join are even lower ranked than themselves. The failure of the producer-scrounger model to predict this pattern likely reflects two of its assumptions. Specifically, the model is built such that any individual playing scrounger can access all the discoveries of others, since (1) resources are assumed to be so rare that they are discovered one-at-a-time and (2) scroungers can access perfect social information and so detect each discovery (*Vickery et al., 1991*; *Giraldeau & Caraco, 2000*; cf. *Ohtsuka & Toquenaga, 2009*). When resources can be monopolised, these conditions mean that only the highest-ranked individuals will benefit from scrounging behaviour. However, in many groups of social foragers multiple patches can be discovered at the same time (and simultaneous discoveries or options will likely be the norm for many other types of resource too). As such, scrounging individuals will be unable to access all resource discoveries even if they have perfect social information (*Ohtsuka & Toquenaga, 2009*; *Afshar & Giraldeau, 2014*). Furthermore, since individuals within social groups are unlikely to be in close proximity at all times (*Krause & Ruxton, 2002*; for this study population, see *Cowlishaw, 1999*; *Castles et al., 2014*), scrounging individuals are unlikely to possess perfect social information regarding all discoveries occurring at the group level (*Barta, Flynn & Giraldeau, 1997*; *Hirsch, 2007*), and any temporal and energetic costs associated with scrounging may vary both within and between individuals.

Under the conditions of simultaneous discoveries and imperfect access to social information, the most dominant individuals will be unable to monopolise all resource discoveries, regardless of the economic defensibility of single resource patches. Instead, as shown here, the difference in rank between individuals across a hierarchy should play an important role in mediating scrounging behaviour. This finding is consistent with the predictions of a recent game theoretic model (*Lee et al., 2016*), which proposes that if the highest ranking animals in a group do not detect a particular patch discovery, and/or are occupied at another discovery, then middle and lower ranking animals can benefit from scrounging, provided that the producers from whom they are scrounging are lower ranking

than themselves. Our result that joining at the sub-patch level still increases with social rank likely reflects the fact that higher ranked individuals have more competitors who are subordinate to them. The most dominant individual should thus be unconstrained in its ability to act upon opportunities to scrounge, and constraints should increase down the dominance hierarchy. Furthermore, theory predicts that if more dominant individuals are more effective at monopolising other's resource discoveries, then they should continue to scrounge even under conditions that drive lower-ranked individuals to switch to the producer tactic (e.g., when the finder's share is large; *Lee et al., 2016*). Individuals may also benefit from positioning themselves so as to maximise scrounging opportunities (*Barta, Flynn & Giraldeau, 1997*; *Di Bitetti & Janson, 2001*; *Hirsch, 2007*). Since dominant individuals may be better able to secure more central positions in the group, this may improve their ability to detect, and increase their proximity to, the discoveries of others (*Barta, Flynn & Giraldeau, 1997*; *King, Isaac & Cowlishaw, 2009*), and further reinforce the effects of dominance on scrounging behaviour.

Second, we showed that the relationship between social dominance and joining is strongly influenced by age class. In stark contrast to adults, juvenile baboons showed no general relationship between social dominance and joining, particularly at the sub-patch level. While an adult baboon's ability to join competitors at this spatial scale depended strongly on their relative dominance, juveniles were less constrained by their social status. Indeed, juvenile behaviour accounted for almost all instances where a subordinate joined a higher-ranking individual at the sub-patch level. There are two likely explanations for this pattern. Firstly, rank acquisition in chacma baboons is mediated primarily through maternal reinforcement. Specifically, a mother will use aggressive behaviours to establish the dominance of her developing offspring over others subordinate to her (*Cheney, 1977*; *Holekamp & Smale, 1991*; *Lea et al., 2014*). Consequently, social rank during early life might be particularly sensitive to context, such that social interactions between juveniles may only reflect rank differences between their mothers in the presence of the dominant mother. In the absence of the dominant mother, older but 'subordinate' juveniles might use their larger size to join younger, smaller, but more 'dominant' competitors. Secondly, there is evidence that juvenile baboons, like the juveniles of several other primate species (*Janson, 1985*), are more frequently tolerated at feeding sites than adults. The presence of co-foraging juveniles may impose only a minimal direct cost to adults, but permitting close kin access to resources may provide inclusive fitness benefits. In particular, father-offspring relationships in chacma baboons afford juveniles access to high-quality feeding sites (*Huchard et al., 2013*). Such toleration may mean that, in addition to better access to monopolisable resources, low-ranked juveniles may not be constrained in their ability to use social information in the same way that similarly ranked adults will be. Our findings do not provide strong evidence in support of either tolerance or juvenile rank instability in disrupting the expected positive relationship between social rank and joining behaviour when resources are monopolisable. However, we might expect juvenile rank stability to disrupt rank asymmetries but still to involve competitive exclusion when resources are monopolisable, while toleration of juveniles should result in reduced levels of competitive exclusion. The fact that the probability of competitive exclusion was reduced for juveniles

compared with adults at the sub-patch scale might suggest that toleration is playing a bigger role than juvenile rank instability in our observations.

Our study demonstrates the way in which the monopolisability of resources may drive social constraints on a subordinate individual's ability to use joining behaviour to access them. We also show that such constraints may be relaxed in juveniles. Since resources generally show some uncertainty in their distribution through space and time, information use is likely to play a key role in resource acquisition. Our study illustrates how competitive processes associated with dominance might facilitate or constrain an individual's ability to benefit from collecting social information when its use requires joining behaviour, as in the producer-scrounger framework (*Barta & Giraldeau, 1998*; *Lee et al., 2016*). An important step in future research will be to develop frameworks that simultaneously consider how resource distributions underpin (1) the strength and type of competition between individuals, (2) the benefits of collecting information socially versus personally, and (3) the rate at which such information becomes out-dated. This approach will elucidate the environmental conditions that should generate interdependencies between contest competition and social information use by highlighting when social information use should be dependent on joining behaviour at a resource. These insights will allow better characterisation of the ways in which competition can modulate relationships between an individual's ability to use information and its access to resources, and the implications of such modulations for the dynamics of natural populations.

## ACKNOWLEDGEMENTS

We would like to say a big thank you to James Ounsley, Cassandra Raby, Rebecca Boulton, Matthis Petit, Eveline Rijksen, Miles Keighley, Maddie Castles, Stef Oberprieler, Alice Baniel, Stella Diamant, Katie Hatton, Julien Collet, Chris Smith, Boris Granovskiy, Caitlin Miller, Alecia Carter, Elise Huchard, Hannah Wilmot, and Willem Odendaal for their work and support in the field. Thanks also to Tim Coulson, Marcus Rowcliffe, Ben Sheldon, David Macdonald, Daniel van der Post, Andrés López-Sepulcre, Alexander Weiss, and three anonymous reviewers for constructive comments and discussions. Permission to work at the field site was kindly granted by Tsaobis Nature Park, the Wittreich and Snyman families, and the Ministry of Lands and Resettlement. We also thank the Gobabeb Training and Research Centre for affiliation in Namibia. This study is a publication of the ZSL Institute of Zoology's Tsaobis Baboon Project.

### Funding

This work was supported by a Natural Environment Research Council Quota Studentship (NE/J500409/1) awarded to AEGL. There was no additional external funding received for this study. The funders had no role in study design, data collection and analysis, decision to publish, or preparation of the manuscript.

## Grant Disclosures

The following grant information was disclosed by the authors:

Natural Environment Research Council Quota Studentship: NE/J500409/1.

## Competing Interests

The authors declare there are no competing interests.

## Author Contributions

- Alexander E.G. Lee conceived and designed the experiments, performed the experiments, analyzed the data, contributed reagents/materials/analysis tools, wrote the paper, prepared figures and/or tables.
- Guy Cowlishaw conceived and designed the experiments, contributed reagents/materials/analysis tools, reviewed drafts of the paper.

## Animal Ethics

The following information was supplied relating to ethical approvals (i.e., approving body and any reference numbers):

Our wholly observational research adhered to the Guidelines for the Use of Animals in Behavioural Research and Teaching (Animal Behaviour 2012 83:301–309), and our protocols were assessed and approved by the Ethics Committee of the Zoological Society of London (BPE/0518).

## Field Study Permissions

The following information was supplied relating to field study approvals (i.e., approving body and any reference numbers):

Our study was approved by the Ministry of Environment and Tourism in Namibia (Research Permits 1696/2012 and 1786/2013).

## Data Availability

All data and code associated with this research are freely available in the Oxford University Research Archive (DOI: 10.5287/bodleian:GPr2aK1ng).

## Supplemental Information

Supplemental information for this article can be found online at http://dx.doi.org/10.7717/peerj.3462#supplemental-information.

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
