# Peer review of "Switching spatial scale reveals dominance-dependent social foraging tactics in a wild primate"

_PeerJ, doi:10.7717/peerj.3462_

## Round 0.1 · original submission · Minor Revisions

Like all three of the experts who I asked to review your manuscript, I was particularly impressed with your article; the data set was large (covering two groups studied over two years), the analyses were thorough and the writing was clear.

Each of the three reviewers have suggested some minor edits that will help to improve the clarity of your manuscript and I only have two additional ones myself:

Line 35: To increase clarity, I suggest providing the reference at the end of the sentence, rather than in the middle.

In addition to group size, could you please provide more detail about the group composition (age, sex etc.) for both of your groups in each of the two seasons.

I look forward to receiving a revised version of your manuscript.

·

Basic reporting

This paper is well-written with clear, professional English used throughout. The Introduction and background provide a good coverage of the literature and provide enough information to understand the paper.

One improvement that should be added is a more precise definition of “social information” early in the Introduction.

The structure conforms to PeerJ standards and discipline norms and raw data is supplied.

Figures and tables are relevant, high quality, well labelled and described.

Experimental design

This paper provides original primary research in the scope of the journal.

The research question is well-defined and there is a clear statement about how this paper fills a gap in the knowledge on producer-scrounger games.

The investigation performed is rigorous with a large data set on a good sample of individuals in a wild setting. The work also shows high technical and ethical standards.

The methods are described in sufficient detail, though I’ve made a couple of small comments below.

Validity of the findings

The data set is large and robust and the authors have done a nice job of choosing the right statistical tests.

The conclusions are linked to both the original research questions and the data analyses. The Discussion is very well-written.

Additional comments

L 31 – This sentence sounds as if social information is associated with low social rank. It needs to be changed to something like, “ability to use social information by those of low social rank”.
L 42 – Add “and knowledge” after “others”.
L 147 – Delete “as each other”.
L 158 – Add “plants” after “conspecifics” to make it clear that you are not talking about monkeys.
L 288 – Change “was” to “were”.
L 294-5 – Change “define” to “defined” and “fall” to “fell”. The authors switch between past and present tense a few times and this is confusing. It is usually best to stay in past tense since the work is already done.
L 296 – A statement here about how distributions were compared is needed. Were these compared statistically? Or just visually?
L 304 – Change “, assessed” to “and were assessed”.
L 313-314 – What alpha level was used to assess significance?
L 363 – Change “on average” to “usually”.
L 460-4 – This sentence is quite long and confusing. Bigger role than what? Consider rephrasing.

Reviewer 2 ·

Basic reporting

This manuscript is very well written. The quality of writing is high, and the manuscript is well structured and easy to follow.

The relevant literature is cited and a good amount of background/context is provided regarding producer scrounger games. There are a three areas that I would recommend expanding:
1) provide the relevant information in the introduction or methods to demonstrate that social foraging dynamics in this study system conform to producer scrounger game and not an information sharing model (i.e. searching for food versus searching for scrounging opportunities are mutuallly exclusive, negative frequency dependent payoffs).
2) provide a more details on the two models of producing/scrounging and dominance with an emphasis on their key assumptions and predictions as relevant to your study design. I elaborate on this point further in the comments to authors section.
3) the predictions regarding different effects of dominance rank on joining probability in juveniles versus adults. There is no mention of this in the introduction. If this was an a priori prediction, I would recommend adding detail regarding age effects in the introduction. If it was post hoc,I would recomment being explicit about this when the age contrasts are introduced in the methods.

The article is structured according to guidelines. The raw data has also been provided and the files contain sufficient detail to allow others to reproduce the results. The authors have also provided their r-scripts, which should be commended.

The manuscript is self-contained and the analyses and follow logically from the predictions.

Experimental design

The study presents original research on the relationship between dominance rank and joining behaviour in socially foraging baboons as a function of resource monopolizability.

The research question is well defined, relevant, and meaningful. The authors clearly describe how the current study fills an existing knowledge gap.

I have no concerns about the technical or ethical standards of the investigation.

The methods are described in sufficient detail for the study to be replicated.

Validity of the findings

The data presented in the manuscript is based on over 800 hours of focal observations producing thousands of observations of joining events in over 100 baboons. The amount of effort to produce this data set is impressive. However, I think it is important to note that these data nonetheless represent observations on only 2 troops of baboons. Given that hierarchies are properties of the group - this really means that there are only 2 independent replicates. I recognize that collecting data on multiple independent troops is likely not feasible, and I believe the data set is valuable even if it comes from only two independent replicates, however, I think the low replication should be explicitly mentioned. One possibility is that rather than combining all data and including troop as a fixed effect, each troop could be viewed as an independent test. If the estimated effects of dominance and monopolizability are the same for both troops, this would provide somewhat greater confidence in the results.

The conclusions presented in the manuscript are clear, directly linked to the original research question, and follow from the results.

Additional comments

The only major comment I have regarding this manuscript is the (mis)alignment of the current analyses with the models that the study aims to test. In particular, the model by the same authors, ‘monopolizability’ of the patch is equated with the inverse of patch area (when c = 0, patch area is large and the patch cannot be monopolized, when c = 100, patch is small and fully monopolizable). The model makes no predictions about the spatial location of producers and scroungers within a patch when scrounging does occur. Instead, it predicts that scrounging should be more common when patches are large, and at smaller patch sizes, the relative rank of individuals should matter (i.e. dominants more likely to scrounge). As such, the construct of patches and subpatches within the current study is not a direct test of the existing models. If possible (i.e. if the authors have data on patch size), it would be useful to reanalyze the data such that the current test is better aligned with the way the model was constructed. If that data is not available, then the differences between the existing models and the current empirical test should be described in more detail, with an emphasis on how and why the misalignment between the model (assumptions and parameterization) and empirical data structure might influence outcomes.

Reviewer 3 ·

Basic reporting

no comment

Experimental design

L115-120: I do not understand why the authors explain what their study is not, rather than simply explaining what it is. If the issue is that it was not possible to determine whether individuals were using a combination of personal and social information when exhibiting joining behaviour, then I feel this should be explicitly mentioned and discussed as such in the Discussion.

Validity of the findings

While I understand the potential issue described by the authors on L276-283, I am confused reading lines 342-354, as my understanding was that the randomization approach would effectively prevent simple increases in the frequency of joining to produce spurious results. The same ambiguity appears on L349-350.

Additional comments

This article reports on an observational dataset of scrounging behaviour in two troops of wild baboons. I found the paper well-written, well-organized, pleasant to read and generally devoid of mistakes. Relevant literature is cited throughout.

Minor comments:
L124: We make
L184: ''although in almost all cases did'': Do you have a value for this? A high % would show the baboons were indeed foraging and not just moving around, and is thus useful to convince the reader these are truly foraging interactions.
L234: these values would be more informative with % included
L235: why include un-weaned individuals, as it is said above their foraging interactions were excluded?
L300: this is again a GLMM as you have random intercepts (L307)
L440: 'particularly': do you mean that this effect is even less significant (both slopes seem close to 0 to me), or that this result is particularly revealing for the sub-patch level, where dominance played a strong role in adults?
Table1: the introduction predicted a significant 3-way interaction; nothing seems to be said about the NS result for this complex interaction in the Discussion (or did you predict 3 significant 2-way interactions?) I would also insert some text on the non-significance & significance of interactions in the Results section to make the link between Intro/methods & Results appear more clearly.

---

## Round 0.2 · accepted · Accept

I believe that you have thoroughly addressed each of the reviewers' comments, as well as my own, and am happy to accept your article for publication in PeerJ.